# Chitosan Microparticles Loaded with New Non-Cytotoxic Isoniazid Derivatives for the Treatment of Tuberculosis: In Vitro and In Vivo Studies

**DOI:** 10.3390/polym14122310

**Published:** 2022-06-07

**Authors:** Ionut Dragostin, Oana-Maria Dragostin, Andreea Teodora Iacob, Maria Dragan, Carmen Lidia Chitescu, Luminita Confederat, Alexandra-Simona Zamfir, Rodica Tatia, Catalina Daniela Stan, Carmen Lacramioara Zamfir

**Affiliations:** 1Research Centre in the Medical-Pharmaceutical Field, Faculty of Medicine and Pharmacy, Dunarea de Jos University, 35 Al. I. Cuza Str., 800017 Galati, Romania; ionut.dragostin@yahoo.com (I.D.); camen.chitescu@ugal.ro (C.L.C.); 2Faculty of Pharmacy, University of Medicine and Pharmacy Grigore T. Popa, 16 Universitatii Str., 700115 Iasi, Romania; andreea.panzariu@umfiasi.ro (A.T.I.); maria.wolszleger@umfiasi.ro (M.D.); catalina.stan@umfiasi.ro (C.D.S.); 3Faculty of Medicine, University of Medicine and Pharmacy Grigore T. Popa, 16 Universitatii Str., 700115 Iasi, Romania; luminita.confederat@yahoo.com; 4Department of Pneumology, Faculty of Medicine, University of Medicine and Pharmacy Grigore T. Popa, 16 Universitatii Str., 700115 Iasi, Romania; simona-zamfir@umfiasi.ro; 5Department of Cellular and Molecular Biology, National Institute of Research and Development for Biological Sciences, 296 Splaiul Independentei, 060031 Bucharest, Romania; rodica.tatia@gmail.com; 6Department of Histology, Faculty of Medicine, University of Medicine and Pharmacy Grigore T. Popa, 16 Universitatii Str., 700115 Iasi, Romania; carmen.zamfir@umfiasi.ro

**Keywords:** encapsulation, microparticles, isoniazid derivatives, chitosan, drug toxicity, biocompatibility, tissue alterations

## Abstract

Lately, in the world of medicine, the use of polymers for the development of innovative therapies seems to be a major concern among researchers. In our case, as a continuation of the research that has been developed so far regarding obtaining new isoniazid (INH) derivatives for tuberculosis treatment, this work aimed to test the ability of the encapsulation method to reduce the toxicity of the drug, isoniazid and its new derivatives. To achieve this goal, the following methods were applied: a structural confirmation of isoniazid derivatives using LC-HRMS/MS; the obtaining of microparticles based on polymeric support; the determination of their loading and biodegradation capacities; in vitro biocompatibility using MTT cell viability assays; and, last but not least, in vivo toxicological screening for the determination of chronic toxicity in laboratory mice, including the performance of a histopathological study and testing for liver enzymes. The results showed a significant reduction in tissue alterations, the disappearance of cell necrosis and microvesicular steatosis areas and lower values of the liver enzymes TGO, TGP and alkaline phosphatase when using encapsulated forms of drugs. In conclusion, the encapsulation of INH and INH derivatives with chitosan had beneficial effects, suggesting a reduction in hepatotoxicity and, therefore, the achievement of the aim of this paper.

## 1. Introduction

In one of our previous research works [1], isoniazid was subjected to a structural modulation process that aimed to synthesize new isonicotinoyl hydrazones, which proved to have an improved pharmacotoxicological profile. As part of an ongoing research process, the present paper focused on the effects of the encapsulation of isoniazid derivatives in polymer matrices (chitosan microparticles).

The benefits of the microencapsulation process include the separation of components, which ensures the prevention of drug incompatibility, and the protection of the substances from acids, alkalis, heat, ultraviolet and oxidant agents, which extends their validity [2]. In this regard, the encapsulation method forms a physical barrier around a sensitive compound to decrease its reactivity with extrinsic factors, exert an important impact on the drug delivery profile and provide new features according to the support material used [3]. This technique can modify the drug release profile as it uses dual pH-responsive, porous, double-layered polymers with opposite charges for controlled release applications, which allows the encapsulated active ingredients to remain active, despite the gastric acidity levels, and ready to be released in the intestines [4]. Among the many benefits, the role of microencapsulation as a tool for increasing the low bioavailability of different compounds must be highlighted, such as polyphenols that are used in the treatment of diabetes [5].

Over time, chitosan has become an interesting material for medical applications and has attracted considerable interest due to its unique combination of properties, such as biocompatibility, biodegradability and low toxicity [6]. Due to these characteristics, chitosan has been assigned a number of applications, either by itself or in combination with other natural polymers, in the food, pharmaceutical, textile, agriculture and cosmetics industries [7]. In addition, the following intrinsic therapeutic effects of chitosan are also known: anticholesteremic, antidiabetic, antimicrobial, antioxidant, antidepressant, antitumor and hemostatic [8]. Microencapsulation based on chitosan is usually used to improve the bioavailability of drugs and obtain their sustained release [9]. The process of obtaining microparticles based on biopolymers, such as chitosan, is different depending on the type of active substance that is embedded; for example, the encapsulation of essential oils through the spray-drying technique is used for *Candida albicans* biofilms [10] while the self-assembled chitosan microparticle formation through the depolymerization of chitosan (using potassium persulphate at 60 °C followed by cooling) is used as a tumor angiogenesis inhibitor [11].

In addition to the existing results that have been presented in the literature, the novelty of this study was the testing of the ability of the encapsulation method to reduce the drug toxicity of isoniazid and its new derivatives, as described in our previous paper [1].

## 2. Materials and Methods

### 2.1. Materials

Chitosan with medium molecular weight (CS MMW, 425 kDa, deacetylation degree of 85%), isoniazid (INH), acetic acid, sodium hydroxide, sodium tripolyphosphate (TPP) and organic solvents (p.a.) were purchased from Sigma-Aldrich, Tokyo, Japan. All solvents and reagents were used without preliminary purification. The synthesis of isoniazid derivatives (INH-a, INH-b and INH-c) in the laboratory was described in a previous study by the same authors [1].

Cell and material culture: A stabilized line of NCTC mouse fibroblasts (929 clones) was used for the biocompatibility analysis and cell morphology analysis. 

The assayed cell cultures from well plates, after 72 h of experimentation, were stained with hematoxylin and eosin (Sigma Corporation, Tokyo, Japan) dyes. The cell morphology was examined and pictures were taken using an inverted microscope (Nikon, Tokyo, Japan) and a photo camera (Zeiss AxioCam MRc 5, Göttingen, Germany).

### 2.2. Methods

#### 2.2.1. Structural Confirmation of Isoniazid Derivatives

As a continuation of the ^1^H-NMR and ATR-FTIR analysis from our previous study [1], in the present work, Q Exactive high-resolution quadrupole Orbitrap LC-HRMS/MS was applied for targeted ion fragmentation (t-MS^2^) for the additional structural confirmation of the compounds.

The analysis was carried out using a Thermo Scientific Dionex Ultimate 3000 UHPLC system, consisting of a pump (Series RS) that was coupled with a column compartment (Series TCC-3000RS) and an autosampler (Series WPS-3000RS). The UHPLC system was controlled by Chromeleon 7.2 software (Thermo Fisher Scientific, Waltham, MA, USA and Dionex Softron GMbH, part of Thermo Fisher Scientific, Bremen, Germany). An ultra-performance Accucore UHPLC Column C18 (150 × 2.1 mm, 2.6 µm; Thermo Scientific) was used. The mobile phase consisted of a mixture of ultra-pure water containing 500 µL L^−1^ of formic acid (pH 2.5) and methanol with 500 µL L^−1^ of formic acid.

A HESI (heated electrospray) ion source was used for the ionization in the positive mode. The ion source parameters were optimized. The full scan HRMS analysis of the compounds was performed using a Q Exactive mass spectrometer. The full scan data in positive mode were acquired at a resolving power of 70,000 FWHM at *m*/*z* 200. A scan range of *m*/*z* 100–1000 Da was chosen, the automatic gain control (AGC) was set at 3e6 and the injection time was set to 200 ms. The scan rate was set at 2 scan s^−1^.

For structural information, the fragmentation events were performed in selective reaction monitoring (SRM) mode and were successively applied to each of the selected compounds at a resolving power of 35,000 FWHM and increasing collision energy (15, 20, 30, 60 and 80 NCE (normalized collision energy)).

The data were processed by the Quan/Qual Browser Xcalibur 2.3 (Thermo Fisher). The mass tolerance window was set to 5 ppm. Mass Frontier 8.0 software (Thermo Scientific, Waltham, MA, USA) was used to generate the fragmentation patterns of the selected compounds for a comparison analysis.

#### 2.2.2. Preparation of Chitosan Microparticles Loaded with Isoniazid Derivatives

The isoniazid derivatives (INH-a, INH-b and INH-c) were loaded into chitosan microparticles for oral administration using the ionic gelation method, according to an adapted procedure from the literature [9] that was previously applied by us [12]. For each batch, 0.5 g of active substance (INH, INH-a, INH-b or INH-c) was suspended in 50 mL of 2% *w*/*v* viscous chitosan solution. Then, 5 mL of the medium viscosity suspension was added dropwise through a syringe needle (18 G) to 5 mL of 2% *w*/*v* TPP solution using gentle stirring. The formed microparticles were left in contact with the crosslinking agent solution for 24 h at room temperature to achieve efficient crosslinking and were subjected to gentle shaking at 300 rpm to prevent their adhesion. Subsequently, the microparticles were extracted from the TPP solution, washed three times with distilled water (to remove excess crosslinking agent) and dried by exposure to the open air (Figure 1).

#### 2.2.3. Microscopic Characterization of the Obtained Microparticles

The size and morphology of the obtained microparticles were analyzed using the Fei Quanta 200F scanning electron microscope (SEM), which allowed for the examination of details with a great magnitude and resolution.

#### 2.2.4. Loading Efficiency (LE)

The loading efficiency of INH derivatives in the chitosan microparticles was evaluated using the UV spectrophotometric method. The content of INH and its derivatives in the TPP solution was evaluated spectrophotometrically at 280 nm, after removing the chitosan microparticles. The loading efficiency (%) was calculated using a calibration curve for each INH derivative and the following formula [13]:% LE (loading efficiency) = (C_0_ − C_1_/C_0_) × 100(1)
where C_0_ is 5 mg/mL (the initial concentration of INH derivative in the chitosan solution that was dripped into the TPP solution) and C_1_ is the concentration of INH derivative in the TPP solution (mg/mL) after removing the formed microparticles.

#### 2.2.5. Assessment of In Vitro Biodegradation Capacity

The biodegradation capacity of the microparticles was studied according to the literature and using techniques that were applied by us in a previous study [14], using lysozyme 152,975 IU/mg as the biodegradation agent after pre-solubilization at 37 °C in a phosphate buffer (pH 7.4) at a concentration of 10,000 IU/mL. Until the balance of swelling was established, the microparticles were left in the buffer solution (pH 7.4) and were then transferred to the lysozyme buffer and kept in the incubator at 37 °C for 7 days. The solution was changed daily. On Days 1, 4 and 7, the microparticles were removed from the buffer medium and weighed, following a decrease in mass due to biodegradation. The evaluation of the biodegradation percentage (*D* %) was calculated using the following formula:(2)$$D\;\%=\frac{W_{0}-W_{x}}{W_{0}\times 100}$$
where *W*_0_ is the mass of the microparticle before incubation and at the equilibrium of its swelling capacity in the buffer solution and *W_x_* is the mass of the microparticle after incubation (on Days 1, 4 and 7).

#### 2.2.6. Evaluation of In Vitro Biocompatibility Using MTT Cell Viability Analysis

An evaluation of the cytotoxicity of the microencapsulated samples was performed by directly exposing cultured cells to samples, followed by a thiazolyl tetrazolium bromide MTT cell proliferation assay.

For this test, the microparticles were sealed in polyethylene foil and sterilized by exposure to UV radiation for 8 h. Cells in 24-well plates were incubated for 24 h at 37 °C and 95% relative humidity in air containing 5% CO_2_ to induce cell adhesion. After 24 h of cell incubation, the medium was replaced with 500 μL of fresh culture medium containing microparticles from each sample. In parallel with the samples, cell cultures treated with H_2_O_2_ hydrogen peroxide (0.03%) were used as a positive control and untreated cells were used as a control culture. The plates were incubated at 37 °C and the quantitative assessment of cytotoxicity was performed after the desired exposure time (24, 48 and 72 h) using MTT thiazolyl tetrazolium bromide. Cell morphology was assessed after 72 h using an inverted microscope. All tests were performed in triplicate.

Cell viability was performed and calculated according to the formula described in the literature [15]:Cell viability (%) = OD test/OD control × 100(3)
where OD test is the optical density of the microparticle sample and OD control is the optical density of the untreated control.

For this, the culture medium was replaced with fresh medium containing an MTT solution in a 10:1 (*v*/*v*) ratio and the plates were incubated at 37 °C for 3 h. Further, a volume of 500 μL of isopropanol was added to each well to dissolve the formazan crystals by gentle shaking on a platform for 3 h. As previously described, the optical density (OD) of the colored solution was read at 570 nm using a Mithras LB940 microplate reader (Berthold Technologies, Bad Wildbad, Germany).

#### 2.2.7. Biological Evaluation

Chronic toxicity studies were performed according to the guidelines for the deontology and ethics of laboratory animal studies (national law no. 206/27 May 2004; directives 2010/63/EU and CE86/609/EEC) and after obtaining approval from the research ethics commission at the University of Medicine and Pharmacy, Grigore T. Popa, Iasi, (17 April 2018). The experimental protocol included the housing and handling of animals and the administration of test compounds in encapsulated form, as well as their scarification after anesthetic administration (ketamine i.p. 100 mg/kg body weight).

##### In Vivo Toxicological Screening: The Determination of Chronic Toxicity

Based on the results obtained from the acute toxicity test (as detailed in our previous study [1]), the chronic toxicity of the isoniazid derivative compounds was studied by the oral administration of the microencapsulated form in doses of 1/10 from DL 50, expressed in mg/kg body weight (1/10 from 175.2 for INH, 352.54 for INH-a, 1778.8 for INH-b and 1251.5 for INH-c).

Thus, the doses calculated based on acute toxicity and used for chronic toxicity were as follows:17.52 mg/kg body weight for isoniazid (INH) in Group 1;35.234 mg/kg body weight for INH-a in Group 2;177.88 mg/kg body weight for INH-b in Group 3;125.15 mg/kg body weight for INH-c in Group 4.

All encapsulated compounds were orally administered daily, in a single dose, for 30 days. The empty chitosan microparticles were used as a control in Group 5, while Group 6 was used as the untreated control batch.

At the end of the experiment, biological products from the blood and liver fragments were collected for biochemical and histopathological analyses. The liver fragments were subsequently fixed in 10% buffered formalin for histopathological examination.

##### Histopathological Examination

In order to evaluate the effects of the microencapsulation of selected compounds on chronic tissue toxicity, a histopathological study was performed on liver tissue fragments to identify possible morphological changes that were produced by the chronic administration of the encapsulated substances. The liver fragments fixed in the 10% buffered formalin were subsequently processed following the specific steps of the histopathological technique [16,17].

The microscopic examination was performed using a Nikon Eclipse 50i microscope to see whether there were any tissue alterations that demonstrated the impact of the administration of the substances incorporated in the chitosan microparticles.

##### Evaluation of Biochemical Parameters

The biochemical parameters of serum aminotransferases (GPT and GOT) and alkaline phosphatase were determined as markers of hepatotoxicity using the ABX Pentra 400 automated biochemical analyzer (Horiba, Kyoto, Japan). The kits were from Diamedix and the reagents were in boxes that ensured their increased stability and the extended linearity of the results.

#### 2.2.8. Statistical Analysis

Data were analyzed using an analysis of variance (ANOVA; *p* < 0.05). All determinations were made in triplicate and the results were expressed as the mean ± standard deviation (SD). Graphical representations were made for all results that referred to the evaluation of the biochemical parameters, which took into account the mean of all animals in each group and showed the standard deviation as error bars.

## 3. Results

### 3.1. Mass Spectrometric Structural Confirmation of the Active Compounds

Recently, high-resolution mass spectrometry (HR-MS/MS) has been applied to the elucidation of the structures of various known and unknown compounds [18,19]. High mass spectral resolving power (RP), which reduces interference, and high mass accuracy, which allows the prediction of molecular formula for spectral peaks, have been proven to be powerful tools for the identification of new chemicals. Structural proposals that are only based on mass spectral evidence are now commonly reported in peer-reviewed literature [18,20,21].

In the present study, a high-resolution full scan analysis and fragmentation experiments for protonated molecules using increasing collision energy were used for compound identification (Figure 1) and the structural characterization of the isoniazid derivatives. Mass Frontier 8.0 software (Thermos Scientific, Waltham, MA, USA) was used to compare the identification of the fragmentation patterns. The fragment ions of isoniazid that were observed in the MS^2^ spectrum of isoniazid and its derivatives are shown in Table 1.

**Figure 1 polymers-14-02310-f001:**
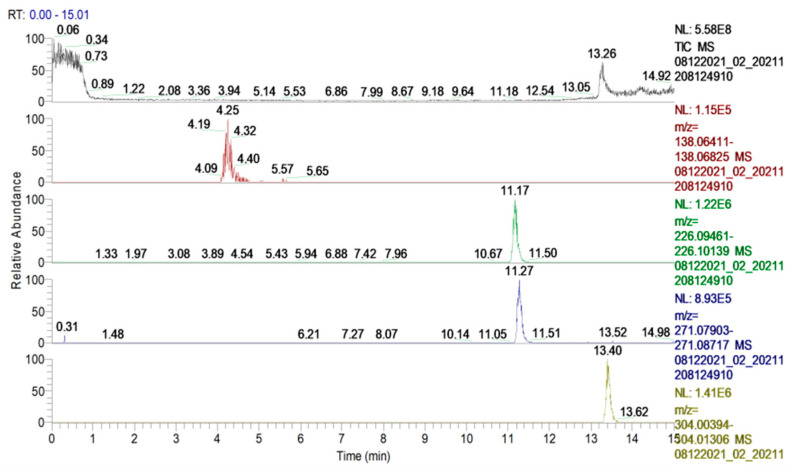
UHPL-HRMS full scan extracted chromatograms of the target compounds (isoniazid and its derivatives). From top to bottom: INH *m*/*z* 138.0667; INH-a *m*/*z* 226.0980; INH-b *m*/*z* 271.0831; INH-c *m*/*z* 304.0085.

The fragment ions *m*/*z* 106.087 and 96.0443, which are both generated by the cleavage of the C–N bond and result in the loss of the [-NH-NH_2_] group, were common for all compounds, which showed the connection to the same parent compound (isoniazid) (Table 1).

For structure confirmation, other fragment ions were identified and compared to the fragmentation patterns that were generated by the Mass Frontier software for all derivatives. In Figure 2, the fragmentation pattern that was identified for INH-c is compared to the parent compound INH as an example.

### 3.2. Macroscopic and Microscopic Characterization of the Obtained Microparticles

The color of the microparticles that contained encapsulated active substances varied from white to yellow, according to substance (Figure 3).

The morphological study showed that the (empty) chitosan microparticles had a regular contour, a spherical and slightly rough surface and a size of about 250 μm in the dry state, showing relative transparency (Figure 4).

The most important SEM characteristics for chitosan microparticles containing active substances were an irregular contour and a smooth, matte and cracked surface, which could be explained by the hydrophobic nature of the encapsulated substances and by their accentuated crystallinity (which was described in our previous study [1]). Their size was in the range of 500–800 μm in the dry state, which demonstrated a significant increase in diameter compared to the empty microparticles that did not contain an active substance (Figure 5).

### 3.3. Loading Efficiency (LE)

The loading efficiency of INH and its derivatives in the chitosan microparticles are shown in Table 2.

As can be observed, the amount of INH derivative that was loaded into the matrix of chitosan was slightly lower than the encapsulated isoniazid (between 89.6 and 92.17% compared to 94.36% for pure isoniazid).

### 3.4. In Vitro Biodegradation Capacity

The results of the in vitro biodegradation using lysozyme, which was monitored over 7 days, are shown in Figure 6.

From Day 3 of incubation, biodegradation occurred more than 20%, ranging from 24.34% (for CS–INH-a microparticles) to 48.49% (for CS–INH-b). Thus, there was a decrease in the biodegradation capacity of the corresponding microparticles due to the increase in the degree of lipophilicity and the occurrence of condensation reactions between isoniazid and different aromatic benzaldehydes. Compared to these microparticles, the empty chitosan microparticles displayed the highest biodegradation ratio of 55.83% (on Day 3). On the last day of incubation (Day 7), the biodegradation of empty microparticles increased to 90.15%, while it increased to a maximum of 75.87% for microparticles containing isoniazid derivatives (for CS–INH-b).

### 3.5. In Vitro Biocompatibility Using MTT Cell Viability Analysis

The cell viability test results are shown in Table 3.

The MTT testing results for the biocompatibility evaluation of the CS–INH microparticles on NCTC cells revealed that microparticles that were based on INH-b and INH-c compounds had manifested biocompatibility with fibroblast cells at all three intervals with viability values of 80.22–106.54% (CS–INH-b at 24 h), which were close to the values that were registered by the negative control, CS–INH and CS. The sample of CS–INH-a microparticles after 24 h was non-cytotoxic, while an increased cytotoxic effect was induced after 48 and 72 h.

The values recorded for the encapsulated derivatives were close to those of the empty chitosan microparticles (CS), i.e., 90.92–98.32%, which were used as the negative control (Table 3).

The influence of the tested microparticles on the cell culture presented in the images indicated a good biocompatibility of samples with NCTC cells, with exception of CS–INH-a, which induced a slightly cytotoxic effect with slow cell proliferation and the modification of the cell shape to an elongated or spindly form (Figure 7).

### 3.6. Biological Evaluation

#### 3.6.1. In Vivo Toxicological Screening: The Determination of Chronic Toxicity

In order to evaluate the impact of the administered encapsulated compounds, on the tissues, liver fragments were examined and the following aspects were highlighted:

Group 1, which received INH microparticles (CS–INH), showed fewer signs of liver damage, manifested only by sinusoidal dilatations and rare inflammatory reactions. Comparing this group to the non-encapsulated INH [1], it could be seen that the areas of microvesicular steatosis disappeared through the encapsulation process, probably due to the hypolipidemic effects of chitosan (as mentioned in the introduction of this paper). Group 2, which received INH-a microparticles (CS–INH-a), also exhibited a pronounced dilation of the sinusoidal capillary system without necrosis but with rare inflammatory infiltrates, while Group 3 (INH-b microparticles) revealed extensive vascular congestion. For Group 4, which received INH-c microparticles, some of the hepatocytes exhibited finely vacuolar cytoplasm. In addition, due to the non-toxic effects of chitosan (which were referred to in the introduction section), Group 5 (CS, non-active empty microparticles) showed normal-looking hepatocytes. The untreated control batch (Group 6) revealed a normal liver morphology, with hepatocyte cords radiating from the centrilobular vein and a normal configuration of liver parenchyma (Figure 8).

#### 3.6.2. Evaluation of Biochemical Parameters

Alanine aminotransferase (ALT)/glutamic pyruvate transaminase (GPT).

A significant reduction in GPT values was observed in the chitosan–INH group (85.75 U/L) and the three encapsulated derivatives: chitosan–INH-a (78.46 U/L) and chitosan–INH-b (76.20 U/L). The lowest values of the GPT were recorded in the group that received INH-c–chitosan (75.15 U/L) and the control group that received empty chitosan microparticles (22.97 U/L), as well as the untreated control group (23.70 U/L) (Figure 9).

Aspartataminotransferase (ASAT)/glutamic oxaloacetic transaminase (TOG).

A slight reduction in GOT values was observed for the encapsulated isoniazid (chitosan–INH; 159.58 U/L) and the three encapsulated derivatives: chitosan–INH-a (147.13 U/L), chitosan–INH-b (145.06 U/L) and chitosan–INH-c (145.18 U/L). The lowest values of the GOT indicator were found in the control group that received empty chitosan microparticles (76.28 U/L) and the untreated control group (74.76 U/L) (Figure 10).

Alkaline phosphatase (ALP).

Alkaline phosphatase values ranged from 175.44 U/L to 182.06 U/L for INH and its encapsulated derivatives. Those values were significantly higher than those recorded in the control group (98.16 U/L) and untreated control group (97.21 U/L) (Figure 11).

## 4. Discussion

Hepatotoxicity may be induced by drugs such as isoniazid, paracetamol, halothane, hydralazine, diclofenac and carbamazepine, which cause hepatotoxic reactions within 1 to 8 weeks in combination with a rash, fever and eosinophilia [22]. One of the main side effects of isoniazid is hepatotoxicity, which is a major cause of treatment discontinuation among tuberculosis patients. Human genetic studies have shown that cytochrome P450 2E1 (CYP2E1) is involved in the metabolism and hepatotoxicity of INH.

As demonstrated in our previous study [1], even when antimicrobial activity decreased with the introduction of attractive electron groups at positions 2 and 4 in the structure of benzaldehyde, higher MIC values in the case of new derivatives (0.84–4.16 μg/mL) were recorded. This happened simultaneously with a significant reduction in the toxicity of the studied compounds of the three isoniazid derivatives. Moreover, their activity was similar or even higher than that of other isoniazid derivatives that have been mentioned in the literature: new thioamides as isoniazid derivatives, with MIC values between 0.391 and 6.25 μg/mL [23]; new isoniazid–azole hybrids, with MIC values between 0.195 and 1.56 μM; and the new isoniazid–pyrrole hybrid, which was found to be active with a MIC value of 3.2 μg/mL [24].

Hepatic toxicity caused by INH can be manifested by the occurrence of cell necrosis, steatosis or both due to the metabolites of the drug, including hydrazine and acetyl hydrazine, having destructive effects on liver cells. In animal experiments, a positive and significant correlation has been reported between plasma hydrazine levels and the severity of liver cell destruction caused by isoniazid treatment [25].

In our previous study [1], we concluded that the least hepatic damage was exhibited by the group of animals that received the INH-c derivative, which showed only slight vascular congestion. Thus, the condensation of isoniazid with bromobenzaldehyde had the most favorable influence on reducing the toxicity of the parent compound at the tissue level.

Microvesicular steatosis was only reported in the group of animals that received isoniazid in suspension, while cell necrosis was completely absent in the case of INH-c but still present in the INH-a and INH-b derivatives cases. Although condensation with bromobenzaldehyde has been proven to be the most favorable, the use of benzaldehyde and nitrobenzaldehyde for the chemical modification of isoniazid presented positive effects in both the reduction in cell necrosis and the disappearance of microvesicular steatosis, while maintaining increased therapeutic potential.

On the other hand, microparticles have been studied as a method for the controlled release of drugs for the pulmonary administration of active substances used to treat various lung diseases, such as asthma, chronic obstructive pulmonary disease or various infectious pathologies [26]. Microencapsulation offers the advantages of protecting the drug against pulmonary metabolism, while ensuring the sustained and prolonged release of the drug. These benefits could be all the more beneficial in the treatment of tuberculosis by increasing patient compliance with long-term therapies, thus allowing for less frequent dosing. Another advantage could be the reduction in the long-term side effects that are associated with systemic or oral tuberculosis therapies [27]. In addition, chitosan has beneficial effects (through its biocompatibility, bioactivity and non-toxicity) on the toxicological and pharmacokinetic profiles of the active substances [28].

Furthermore, during the microencapsulation process, the size of the microparticles is easily controllable by setting the necessary parameters, which is a plus in that the resulting particles can efficiently penetrate into the alveoli due to their aerodynamic shape. Thus, higher intracellular drug concentrations in macrophages could be obtained [29].

There are various studies in the literature that have aimed to encapsulate active principles that are used in the treatment of tuberculosis using biodegradable polymers, such as polylactic acid (PLA) [30], poly (lactic-co-glycolic acid) (PLGA) [31], hyaluronic acid [32] or chitosan [33,34]. Among the substances included in the microencapsulation process, both first-line drugs that are used in the treatment of tuberculosis (such as rifampicin, isoniazid and pyrazinamide) and second- or third-line drugs (such as para-aminosalicylic acid (PAS), capreomycin, amikacin and moxifloxacin) have been considered [35,36,37].

In one study [38], a dose–effect relationship was observed between poly (lactic-co-glycolic acid) microspheres that contained rifampicin and pulmonary bacterial loading, after either single or repeated administration. In addition, guinea pigs that were treated with these microspheres had a significantly lower number of viable bacteria but reduced inflammation and lung damage compared to control animals.

In another study presenting the encapsulation of rifampicin in chitosan, the resulting microparticles showed a significant increase in the release of the active substance compared to other polymers [39]. The same authors encapsulated isoniazid in chitosan microparticles and found that their size allowed release to the lower airways, but they did not include animal studies to test the benefits to the treatment of tuberculosis.

In our study, the influence of microencapsulation on cell biocompatibility was tested in vitro and the influence on hepatotoxicity was tested in vivo. An improvement in cytotoxicity was observed for the drugs that were encapsulated in chitosan compared to the non-encapsulated versions, which was confirmed by higher values of cell viability percentage.

For example, INH-c, the derivative with the highest cell viability after 72 h, had a cell viability percentage of 94.14% in its encapsulated form, while the percentage was 72.47% in its non-encapsulated form [1].

The encapsulation of isoniazid derivatives using chitosan as a microencapsulation polymer displayed beneficial effects on chronic toxicity reduction. Hepatocellular lesions caused by necrosis and microvesicular steatosis were only observed in animals that received pure isoniazid during the 30-day experiment [1].

These abnormalities were reduced by administering microencapsulated treatment, both in terms of the reduction in areas of cell necrosis and the disappearance of microvesicular steatosis. Similar effects were observed for all isoniazid derivatives in encapsulated form, in contrast to non-encapsulated compounds.

Thus, the importance of using chitosan to reduce isoniazid hepatotoxicity was particularly emphasized. Unlike non-encapsulated isoniazid INH [1], the encapsulated form CS–INH resulted in reduced areas of advanced cell necrosis as well as the disappearance of microvesicular steatosis. The lipid-lowering effects of chitosan, which are known in the literature [40], had a positive influence on microvesicular stasis. Moreover, the hepatoprotective effects of chitosan could be explained by its antioxidant action, as mentioned in various studies [41,42,43,44].

Regarding the encapsulated form of INH-a (in contrast to the non-encapsulated version for which areas of the onset of necrosis were described), the disappearance of areas of cell necrosis was noticed, which was similar to the administration of INH-b microcapsules. In the case of INH-c, the non-encapsulated form did not produce significant liver damage and no changes in tissue morphology were reported for the encapsulated form.

The hepatotoxicity analysis of laboratory animals was also evaluated by the determination of liver enzyme activity. Thus, serum GPT and GOT levels (markers for liver damage) increased significantly in animals that received parental isoniazid, indicating hepatotoxicity due to the oxidative destruction of hepatocytes, which resulted in the release of enzymes into the vascular compartments. The hepatotoxicity was significantly diminished by the administration of drugs in a microencapsulated form due to the intrinsic properties of chitosan and the structural changes that were made to the isoniazid molecule, which were addressed in the introduction section. Thus, in the case of groups that received encapsulated substances, there was a decrease in the values expressing the activity of liver enzymes compared to the groups that received non-encapsulated substances. This was closely related to the disappearance of cell necrosis areas and microvesicular steatosis that was highlighted during the histopathological examination, which confirmed the correlation of the data obtained from the two types of analysis.

In this context, the use of chitosan microencapsulation for tuberculostatic therapies could represent an important strategy for reducing liver enzyme levels, removing microvesicular steatosis and reducing cell necrosis areas. All of this would have a very important contribution to reducing the risk of fulminant liver failure.

## 5. Conclusions

Hepatotoxicity induced by tuberculostatic therapy is a significant adverse effect that interferes with the effective administration of tuberculosis treatment. In this regard, our study presented the beneficial effects of the microencapsulation of specific drugs on liver damage induced by anti-TB therapy. The chitosan microparticles loaded with isoniazid derivatives that were subjected to our research proved to be an advantageous formulation for reducing the hepatotoxicity of the studied compounds and ensuring the disappearance of cell necrosis and microvesicular steatosis compared to their administration in suspension. In conclusion, the microparticles obtained in innovative pharmaceutical forms have potential biological application as antimicrobial agents in the treatment of tuberculosis, for which both a favorable antimicrobial action and a low incidence of adverse reactions are required given the problem of high toxicity caused by current tuberculostatic medication.

## Figures and Tables

**Figure 2 polymers-14-02310-f002:**
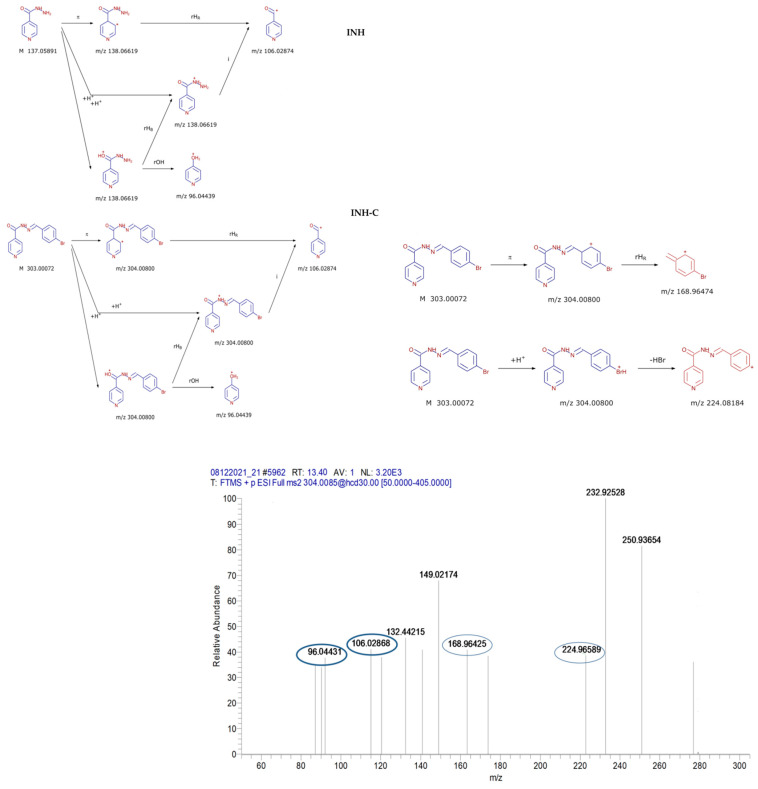
Common fragmentation pathways of INH-c and the parent compound INH and the characteristic fragments of the derivative compound identified in the MS^2^ spectra.

**Figure 3 polymers-14-02310-f003:**
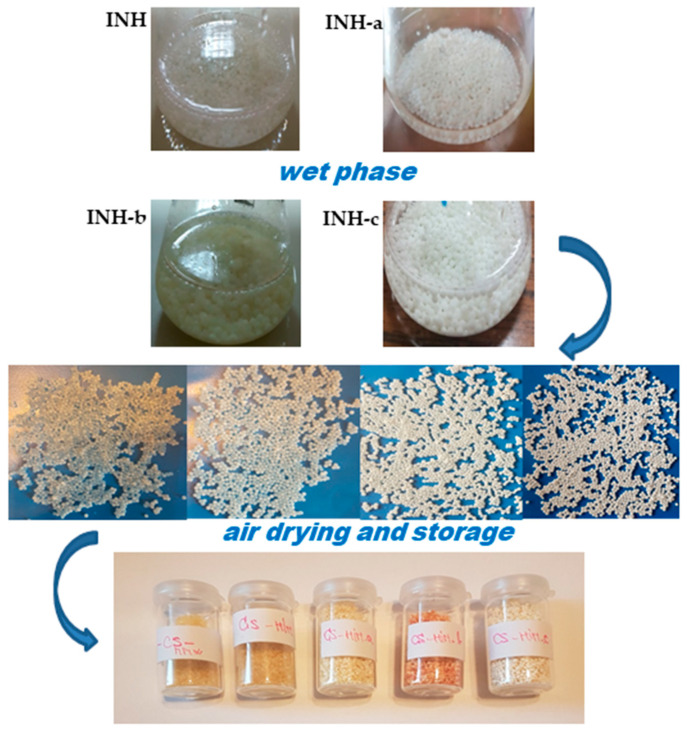
Obtaining chitosan microparticles containing isoniazid (INH) and its condensation products (INH-a, INH-b and INH-c): the wet phase and air drying.

**Figure 4 polymers-14-02310-f004:**
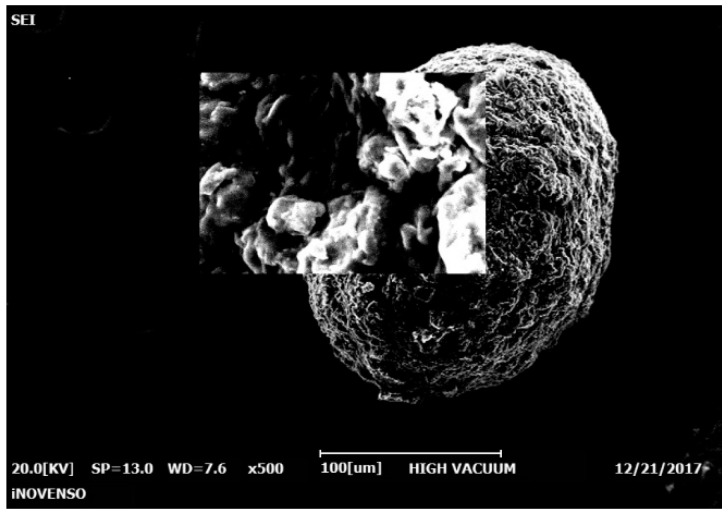
SEM image of an empty chitosan microparticle.

**Figure 5 polymers-14-02310-f005:**
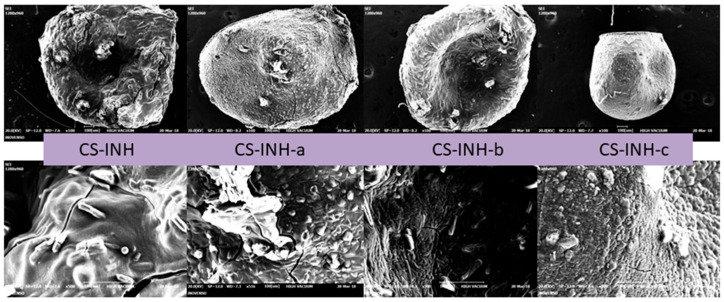
SEM image of chitosan microparticles containing active substances (CS–INH, CS–INH-a, CS–INH-b and CS–INH-c).

**Figure 6 polymers-14-02310-f006:**
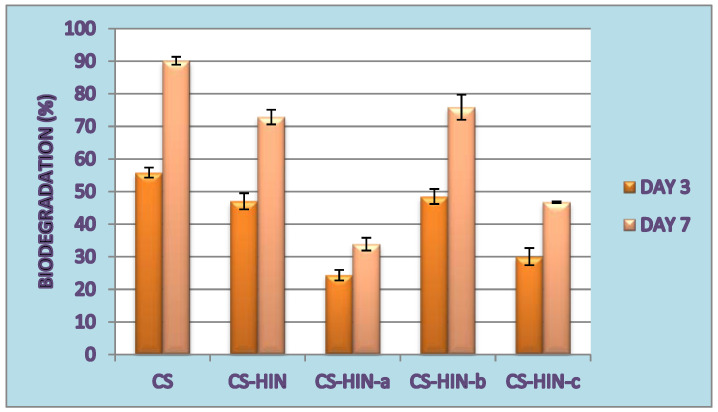
In vitro biodegradability of microparticles containing active substances compared to that of empty microparticles (CS).

**Figure 7 polymers-14-02310-f007:**
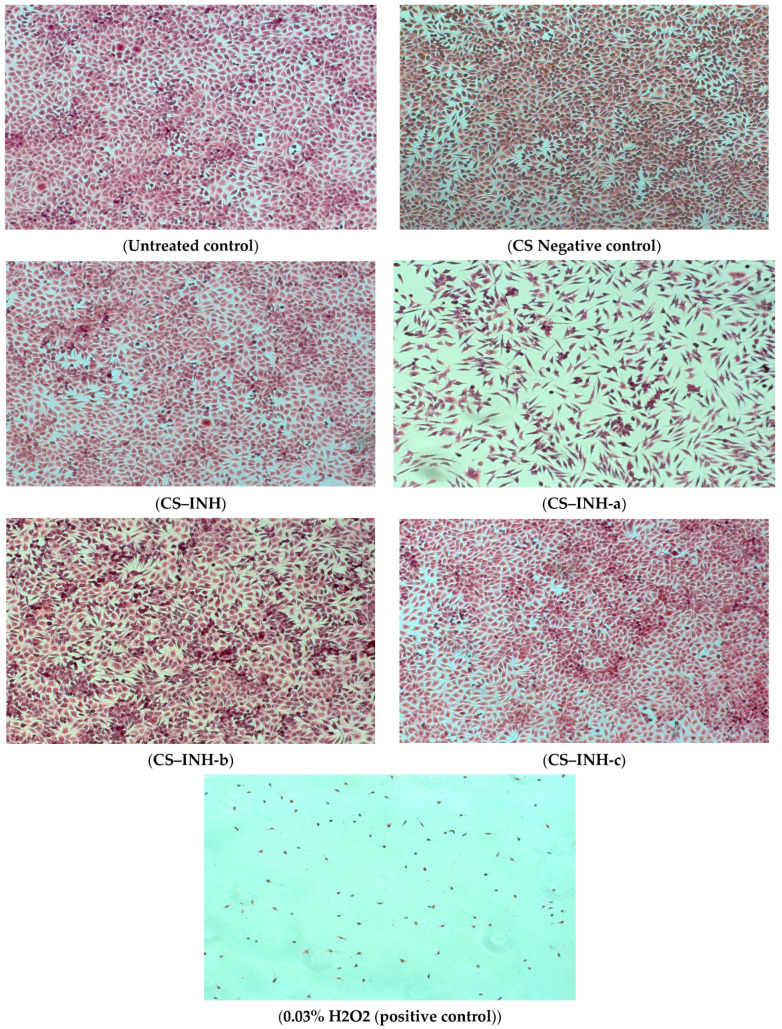
Cell morphology stained with hematoxylin–eosin after 72 h of incubation with microparticles of CS–INH derivatives, CS and CS–INH negative controls.

**Figure 8 polymers-14-02310-f008:**
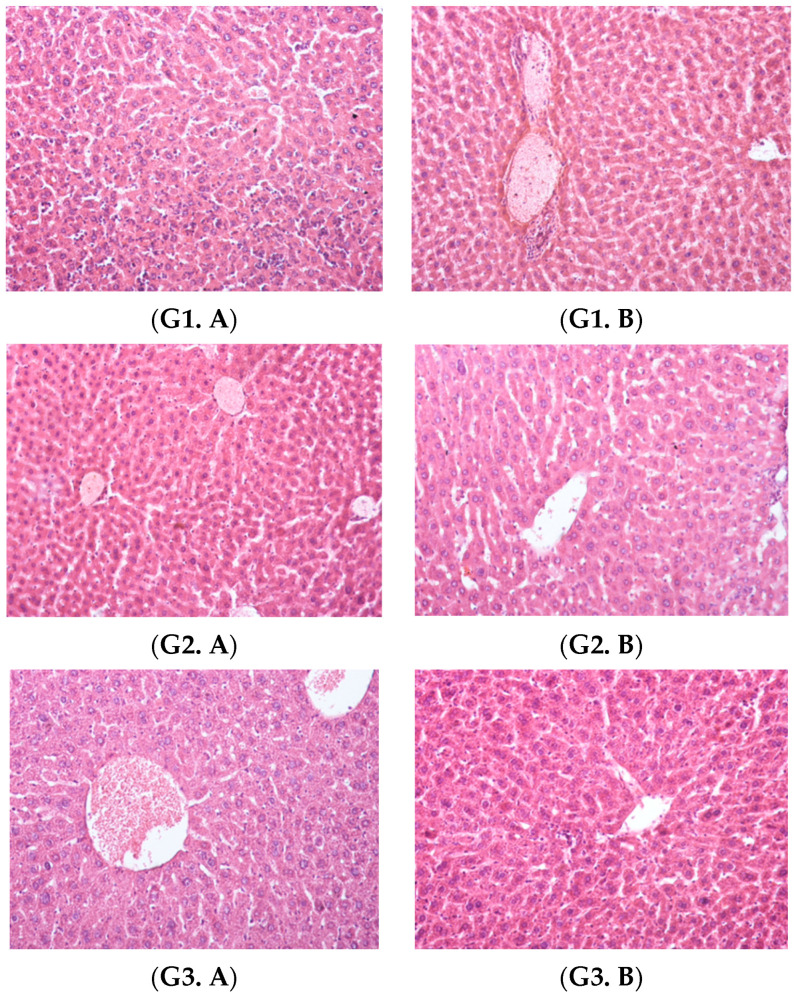
Liver, hematoxylin and eosin staining × 20: Group 1, (**G1. A**) sinusoidal dilatations, capillary mononuclear cell infiltrations and the onset of necrosis and (**G1. B**) small areas of perivascular inflammation; Group 2, (**G2. A**) areas of vascular congestion and (**G2. B**) sinusoidal dilatations and reduced areas of inflammation; Group 3, (**G3. A**) extensive vascular congestion and (**G3. B**) slight sinusoidal dilatations; Group 4 (**G4**), hepatocellular vacuolation and reduced vascular congestion; Group 5 (**G5**), normal liver morphology; Group 6 (**G6**), normal liver morphology.

**Figure 9 polymers-14-02310-f009:**
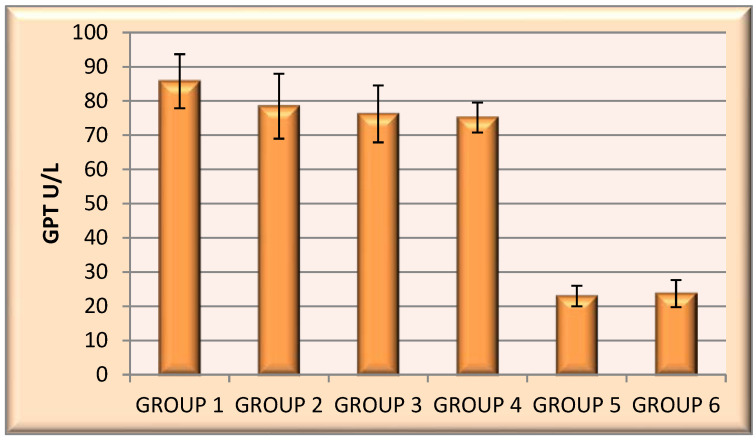
Mean GPT values for the encapsulated samples from the six groups of animals.

**Figure 10 polymers-14-02310-f010:**
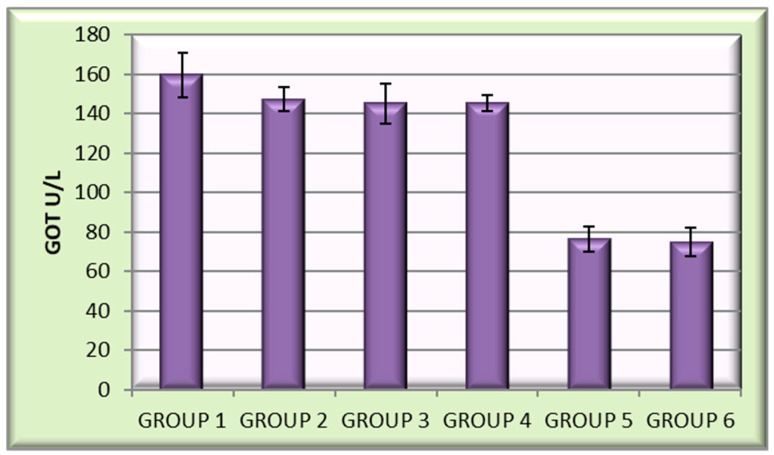
Mean GOT values for the encapsulated samples from the six groups of animals.

**Figure 11 polymers-14-02310-f011:**
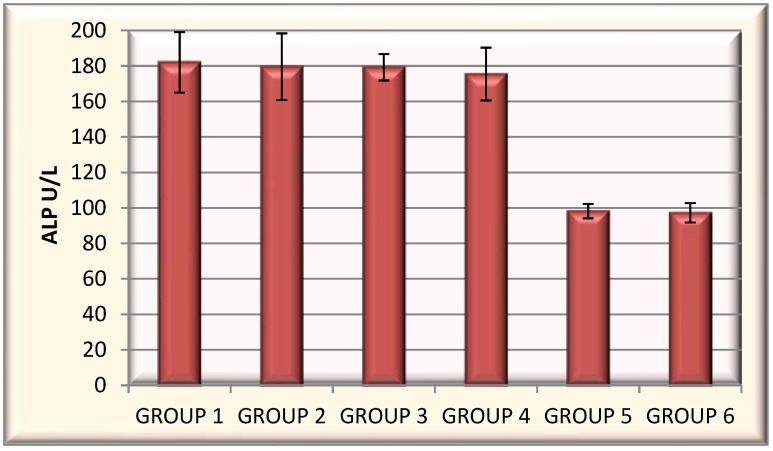
Mean alkaline phosphatase values for the encapsulated samples from the six groups of animals.

**Table 1 polymers-14-02310-t001:** The target compounds, deprotonated precursors and fragment ions. The common fragments for the parent compound and its derivatives are in bold.

Compound	R.T. (min)	Formula	Exact Mass	Error (ppm)	Adduct Ion (*m*/*z*)	MS^2^ Fragments (*m*/*z*)
INH	4.25	C_6_H_7_N_3_O	137.0589	1.47	138.0667	122.0712; 120.0556; 108.04429; 106.0287; 96.0443; 59.0239
INH-a	11.17	C_13_H_11_N_3_O	225.0902	1.25	226.0980	208.08692; 148.0505; 147.05527; 106.0287; 96.0443
INH-b	11.27	C_13_H_10_N_4_O_3_	270.0753	0.86	271.0831	255.0876; 253.0720; 224.0818; 192.0430; 148.0505; 136.0393; 106.0287; 96.0443
INH-c	13.40	C_13_H_10_N_3_OBr	303.0007	0.5	304.0085	288.0130; 285.9974;224.9658; 224.0818; 168.9647; 148.0505; 106.0287; 96.0443

**Table 2 polymers-14-02310-t002:** Loading efficiency (LE) of chitosan microparticles.

Compound	C_0_ (mg/mL)	C_1_ (mg/mL)	LE (%), *n* = 3
CS–INH	5	0.282	94.36 ± 0.7
CS–INH-a	5	0.52	89.60 ± 0.9
CS–INH-b	5	0.39	92.17 ± 0.8
CS–INH-c	5	0.45	90.89 ± 0.7

**Table 3 polymers-14-02310-t003:** The cell viability values (%) for the tested samples of powder compounds and microparticles after 24 h, 48 h and 72 h.

Samples	Cell Viability 24 h	(%) 48 h	72 h
CS–INH-a	83.92 ± 0.7	71.08 ± 0.9	63.99 ± 0.6
CS–INH-b	106.54 ± 0.9	99.09 ± 1.2	89.64 ± 0.8
CS–INH-c	94.96 ± 0.5	80.22 ± 0.9	94.14 ± 0.7
CS–INH	100.27 ± 1.1	98.37 ± 0.7	95.53 ± 0.9
CS (Negative Control)	98.23 ± 0.8	90.92 ± 0.4	97.10 ± 0.6
H_2_O_2_ 0.03% (Positive Control)	11.99 ± 0.8	5.02 ± 0.5	2.90 ± 1.4
Untreated Control	100.00	100.00	100.00

80–100%, non-cytotoxic compounds; 50–80%, mildly cytotoxic compounds; 30–50%, moderately cytotoxic compounds; <30%, severely cytotoxic compounds (ISO 10993-5, Geneva 2003).

## Data Availability

Not applicable.

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
