# Peer review of "Chitosan Microparticles Loaded with New Non-Cytotoxic Isoniazid Derivatives for the Treatment of Tuberculosis: In Vitro and In Vivo Studies"

_polymers, 2022, doi:10.3390/polym14122310_

Round 1

Reviewer 1 Report

Minor remarks

Avoid the use of the first-person plural (for instance, we do that). The scientific manuscript should be written using the third-person singular.

Celsius degree is not adequate and should be retyped. Using symbol from MS office.

Provide a blank space between quantity and unit, only in the case of percentage.

Latin terms should be presented in italics. Check also the references list.

Use the uppercase letters for Figures and Tables.

Use the SI units. For instance, hours, etc. Also, use the following unit for milliliters “mL”.

All equations should be numbered.

All other minor remarks are highlighted in the manuscript.

Major remarks

The following relevant reference in this scientific field should be included in the reference list: DOI: https://doi.org/10.3390/antiox11020297

The novelty of the paper should be mentioned.

Reviewer 2 Report

This study evaluated the prospective of microencapsulation to reduce the toxicity of new isoniazid derivatives namely, isonicotinoyl hydrazones. The authors evaluated the formulation both in vitro and in vivo. The results look promising and the data showed a reduction in the hepatotoxicity due to microencapsulation using chitosan. There are some flaws and the description of the overall story could be improved.

Comments

1.      The major concern I have is regarding the structural confirmation of isoniazid derivatives, is this data already presented in the earlier article? If so, the authors have to delete this section and refer to the earlier study.

2.      Why the characterization was limited to surface morphology and drug loading? The addition of drug release data could have been interesting, though not mandatory.

3.      Is the formulation intended for oral or parenteral? This information is not clear and should be mentioned.

4.      The clinical perspective of this study needs to be emphasized.

5.      A flow cytometry analysis diagram could have been incorporated.

6.      Statistical analysis has to be indicated in the text and images.
